# Membrane Localized GbTMEM214s Participate in Modulating Cotton Resistance to Verticillium Wilt

**DOI:** 10.3390/plants11182342

**Published:** 2022-09-08

**Authors:** Jun Zhao, Jianwen Xu, Yueping Wang, Jianguang Liu, Chengguang Dong, Liang Zhao, Nijiang Ai, Zhenzhen Xu, Qi Guo, Guoli Feng, Peng Xu, Junling Cheng, Xin Wang, Juan Wang, Songhua Xiao

**Affiliations:** 1Key Laboratory of Cotton and Rapeseed, Institute of Industrial Crops, Ministry of Agriculture, Jiangsu Academy of Agricultural Sciences, Nanjing 210014, China; 2College of Agricultural, Xinjiang Agricultural University, Urumqi 830052, China; 3Cotton Research Institute, Xinjiang Academy of Agricultural and Reclamation Science, Shihezi 832000, China; 4Shihezi Agricultural Science Research Institute, Shihezi 832000, China

**Keywords:** Verticillium wilt, cotton, transmembrane protein, resistance, plant immunity

## Abstract

Verticillium wilt (VW) is a soil-borne fungal disease caused by *Verticillium dahliae* Kleb, which leads to serious damage to cotton production annually in the world. In our previous study, a *transmembrane protein 214 protein* (*TMEM214*) gene associated with VW resistance was map-based cloned from *Gossypium barbadense* (*G. barbadense*). TMEM214 proteins are a kind of transmembrane protein, but their function in plants is rarely studied. To reveal the function of *TMEM214s* in VW resistance, all six *TMEM214s* were cloned from *G. barbadense* in this study. These genes were named as *GbTMEM214-1_A/D*, *GbTMEM214-4_A/D* and *GbTMEM214-7_A/D*, according to their location on the chromosomes. The encoded proteins are all located on the cell membrane. *TMEM214* genes were all induced with *Verticillium dahliae* inoculation and showed significant differences between resistant and susceptible varieties, but the expression patterns of *GbTMEM214s* under different hormone treatments were significantly different. Virus-induced gene silencing analysis showed the resistance to VW of *GbTMEM214s*-silenced lines decreased significantly, which further proves the important role of *GbTMEM214s* in the resistance to *Verticillium dahliae*. Our study provides an insight into the involvement of *GbTMEM214s* in VW resistance, which was helpful to better understand the disease-resistance mechanism of plants.

## 1. Introduction

Cotton production is seriously affected by Verticillium wilt (VW), and the breeding of resistant varieties is a major problem faced by breeders. The main reason for this dilemma is the lack of immunity or high resistance in cultivated varieties to VW. With the improvement of biotechnology, transgenic technology is an important way to breed VW resistant varieties, and the core of this technology is to clone important VW-resistant genes. In our previous study, a VW resistance-related transmembrane protein gene, named as *GbTMEM*, was cloned from *Gossypium barbadense* (*G. barbadense*) using map-based cloning [1]. As the results showed, *GbTMEM* was highly expressed in *G. barbadense* when infected with VW, and its silencing expression would reduce disease resistance.

The resistance mechanism of cotton to VW is a very complex process, and research progress is delayed by the lack of cotton germplasm resources immune to VW. With the development of biotechnology, various omics and high-throughput sequencing technologies have been applied to study the resistance mechanism of cotton to VW, and a large amount of valuable data was obtained [2,3]. In addition, virus-induced gene silencing (VIGS) technology is widely used in the study of the cotton resistance mechanism. The function of resistance genes can be confirmed by specific silencing and the observation of phenotypic changes [4,5]. These methods provide an important reference for screening candidate resistance genes and play an important role in elucidating the regulatory network of cotton resistance.

The resistance mechanism of cotton to VW is a very complex biological process, involving a variety of substances and signaling pathways. Terpenoid aldehydes and phenylpropanoids, as well as reactive oxygen species, salicylic acid, jasmonic acid, ethylene, brassinosteroids and other signaling pathways are involved in cotton resistance to VW [4,6,7,8,9,10]. In the process of exploring the resistance mechanism of plants to VW, a large number of genes related to VW resistance were cloned. *Ve1* in tomatoes is one of the well-known resistance genes to VW. But studies have shown that this gene is not involved in the resistance of cotton to VW, which indicates the different resistance mechanisms between cotton and tomatoes [11]. Some *Ve* genes were also cloned in cotton, and the transgenic plants showed a certain degree of resistance to VW. However, further exploration of their resistance mechanisms has not been reported [12,13]. In addition to *Ve* genes, a large number of genes related to VW resistance were cloned from the cotton genome, including *GbRLK*, *GhPAO*, *Gbvdr*, *GbTLP1*, *GbERF1-like*, *GhSAMDC*, *GbNRX1*, *GhLMM* etc. [10,14,15,16,17,18,19]. In addition, some exogenous genes can also improve the resistance of cotton to VW, such as *Hpa1Xoo*, *p35*, *NaD1*, *GAFPs* and *Hcm1* [20,21,22,23]. In this study, we found that a new class of genes, TMEMs, also play an important role in the resistance of cotton to VW.

Transmembrane proteins are a kind of special proteins located in the phospholipid bilayer, which mainly undertake the function of intracellular and extracellular environmental information and material exchange, and are indispensable in biological activities. In the process of plant disease resistance, transmembrane proteins can recognize and accept pathogen signals, activate intracellular reactions, transmit extracellular signals to the intracellular, and induce defense responses. A large number of resistance-related transmembrane protein genes have been cloned in plants, such as *Xa21*, *Pi-d2*, *FLS2*, *Ve1* and *PigmR* [24,25,26,27]. However, the resistance function of *TMEMs* in plants has never been reported. A series of studies showed that the TMEM protein family in animals is involved in intercellular signal transduction, immune-related diseases and tumor development [28,29]. TMEMs have been proved to participate in many physiological processes, such as the formation of plasma membrane ion channels, the activation of signal transduction pathways, mediating apoptosis and autophagy [29]. However, the related research is still in its infancy, and the functions of these proteins have not been fully revealed. In this study, other members of the TMEM gene family in *G. barbadense* were cloned. VIGS and expression analysis showed that these genes were also involved in the resistance of cotton to VW.

## 2. Results

### 2.1. Identification and Phylogenetic Analysis of GbTMEM214s

In our previous study, a VW-resistant gene, *GbTMEM214,* on chromosome D4 of *G. barbadense* was identified with QTL mapping [1]. The gene was located on chromosome D4, and its resistance function was verified with qRT and VIGS experiments. Five other genes in the *TMEM214* gene family were cloned from chromosomes A1, A4, A7, D1 and D7 in this study. The three genes were respectively named *GbTMEM214-1_A/D*, *GbTMEM214-4_A/D* and *GbTMEM214-7_A/D*, according to the chromosomes they are located on. 

A comparison of gene structures revealed differences among the homologs of *GbTMEM214s* (Figure 1 and Appendix A). The structures of *GbTMEM214-1* homologous genes in A- and D- genomes of *G. hirsutum* and *G. barbadense* were the same. The homologous genes of *GbTMEM214-4* were different in the first exon, and similar differences existed among homologous genes of *GbTMEM214-7*. To explore the existence and evolutionary relationship of the TMEM214 protein in plants, proteins encoded with the *TMEM214* homologous genes in *G. barbadense* were further subjected to phylogenetic tree analysis together with homologous genes in other plants. It was shown that the genetic relationship of TMEM214 homologous proteins between the A- and D- genomes was very close (Figure 2). Among TMEM214s in selected plants, GbTMEM214-4 was closer to EOY01031 and EOY01030 in cocoa, while GbTMEM214-7 was closer to EOY03934. 

### 2.2. Protein Structure and Subcellular Localization of GbTMEM214s

To understand *TMEM214s* in cotton, bioinformatic and molecular biologic experiments were carried out to analyze the characteristics of GbTMEM214s. As shown in the results, the homologous proteins of GbTMEM214s in the A- and D- genomes were highly similar in secondary (Figure 3 and Appendix A). Protein domain analysis showed that all six GbTMEM214s contained a TMEM214 domain of different sizes (Figure 3a). The TMEM214 domain of GbTMEM214-7 was significantly shorter than other GbTMEM214 proteins. At the N-terminal of the protein sequence, GbTMEM214-4 and GbTMEM214-7 contained one and two low-complexity regions, respectively. Although the six GbTMEM214s were classified in the TMEM214 protein family, the 3D modeling results showed the wide difference among the 3D model structures (Figure 3b). Although no transmembrane region was found in the protein sequences and the 3D structures were quite different, the three proteins were co-localized with PIP2A on the cell membrane as subcellular localization results showed (Figure 4).

### 2.3. Verticillium Dahliae Induced Expression Analysis of TMEM214s in Cotton

In order to find out the differential expression of *TMEM214s* in susceptible and resistant cotton varieties infected with *Verticillium dahliae* (*V. dahliae*), qRT-PCR was applied to examine the expression of homologs in TM-1 (susceptible variety) and Hai7124 (resistant variety) (Figure 5). As shown in the results, all three *TMEM214s* were induced with *V. dahliae* inoculation and involved in the response to *V. dahliae*. The expression level of *GbTMEM214-4* in Hai7124 was more significantly up-regulated than *GhTMEM214-4* in TM-1 at 24 h and 48 h after inoculation, reaching 5.57-fold and 19.09-fold, respectively. However, the expression of *GbTMEM214-1* and *GbTMEM214-7* in Hai7124 was less up-regulated than their homologous genes in TM-1, *GhTMEM214-1* and *GhTMEM214-7*, from 48 h to 144 h after inoculation, and the expression of *GhTMEM214-7* was significantly increased by 6.00, 1.85 and 3.16-fold, respectively. Overall, the expression of *TMEM214-4* in the resistant variety was more significantly up-regulated than that in the susceptible variety, while the expression patterns of *TMEM214-1* and *TMEM214-7* in the resistant and susceptible varieties were not significantly different.

### 2.4. Phytohormone Induced Expression Analysis of GbTMEM214s

Phytohormones play an important regulatory role in cotton disease resistance. Therefore, their regulatory characteristics on *GbTMEM214s* expression were explored in this study. Under phytohormone induction, the expression patterns of *GbTMEM214s* were quite different. Following treatment, *GbTMEM214-4* was induced with jasmonic acid (JA), salicylic acid (SA), gibberellin (GA) and ethylene (ET), and reduced with indole-3-acetic acid (IAA) and Zeatin. *GbTMEM214-1* was induced with JA, SA, abscisic acid (ABA), GA, Zeatin and ET, and reduced with IAA and brassinosteroid (BR). *GbTMEM214-7* was induced with JA, SA, IAA, GA, and ET, and reduced with ABA, BR and Zeatin (Figure 6). Under ET treatment, *GbTMEM214-4* was highest up-regulated with a maximum of 8.42-fold at 10 h, but the expressions of *GbTMEM214-1* and *GbTMEM214-7* were not so obvious. Among the eight hormones, *GbTMEM214-1* was highest up-regulated with JA with a maximum of 5.48-fold at 1 h, and *GbTMEM214-7* was highest up-regulated with IAA with a maximum of 4.00-fold at 1 h. Three *GbTMEM214s* were not induced with BR, and the expression levels even decreased at 12 h after treatment.

### 2.5. Resistance Function Analysis of GbTMEM214s

To verify the resistance function, *GbTMEM214s* was silenced using VIGS to define loss-function in response to the pathogen. Two weeks after VIGS infiltration, the positive control, TRV:*GbCLA*, showed an obvious photobleaching phenotype, and the expression of *GbTMEM214s* in silenced plants was significantly down-regulated in the corresponding lines (Figure 7a,c). After inoculation with *V. dahliae*, all three *GbTMEM214s*-silenced lines exhibited more wilting, etiolated and even abscission of leaves than mock (Figure 7b and Appendix A). The disease index was calculated 28 days after inoculation, which reached 78.8%, 70.8% and 75.0% respectively for *GbTMEM214-4*, *GbTMEM214-1* and *GbTMEM214-7* deficient lines, compared with 39.3% for Hai7124 and 41.7% for the pTRV2:00 line (Figure 7d). The increased susceptibility of silenced lines indicated the important role of *GbTMEM214s* in cotton resistance to *V. dahliae*.

## 3. Discussion

Transmembrane proteins are a class of proteins with a unique structure, which are ubiquitous in various animal and plant cells. According to plant genome data, 20% to 30% of the proteins have transmembrane domains, indicating that these proteins play a very important and extensive role [30]. 

Based on a previously cloned disease-resistance gene, *GbTMEM214*, the function of its homologous gene in *G*. *barbadense* against VW was analyzed in this study. The results showed that these genes played an important role in cotton disease resistance. A plant’s immune system is composed of cell surface and intracellular immunity [31]. In cell surface immunity, immune receptors sense common signatures of pathogens outside the host cell via extracellular domains (ECDs) and initiate cellular responses to resist infection via intracellular kinase domains (KDs) [32]. Membrane-localized receptor-like kinases (RLKs) and receptor-like proteins (RLPs) are two major components of cell-surface immunity to detect signatures of infection [33]. RLKs contain a variable extracellular domain mediating pathogen recognition, a single-pass transmembrane domain and an intracellular KD-transducing signal to downstream immune pathways [34]. Whereas, RLPs exhibit a similar overall structure to RLKs, but only contain a short intracellular tail, lack a kinase domain, and require a co-receptor to transduce signals [35,36]. Cell-surface immune receptors, also known as pattern recognition receptors (PRRs), monitor the extracellular environment for pathogen invasion patterns, including microbial-associated molecular patterns (MAMPs) and damage-associated molecular patterns (DAMPs) [37,38]. In the process from sensing patterns to immune responses, co-receptors are required to transduce immune signals [39,40]. In this study, GbTMEM214s were found to be located on the cell membrane of plants, and they were considered to play an important role in cell surface immunity. Based on the analysis of the sequence and structure of the three GbTMEM214 proteins, they were obviously different in the secondary and 3D structures. Therefore, the members of this protein family were speculated to play different roles in the resistance of cotton to VW, but the specific mechanism remained to be further studied.

*TMEMs* belong to a large gene family containing the TMEM domain, but their functions are rarely studied and even not reported in plants. According to existing functional studies, it was found that *TMEMs* in animals are related to intracellular signal transduction, immune-related diseases and tumorigenesis, but the function of most genes in this family is still unclear [41]. The immune system of plants is similar to the innate immune system of animals [42]. However, as plants lack an adaptive immune system, they rely solely on natural immunity against pathogens. As reported, several TMEM proteins in animals were found to be involved in immune-related diseases. TMEM9B can activate NF-κB pathway-induced apoptosis, and act as an important factor in the TNF-activated MAPK signaling pathway [43,44]. TMEM176 is related to transplantation immunity, and its overexpression can inhibit rejection after transplantation [45]. TMEM214 in animals, which belongs to the same subfamily as GbTMEM214 in this study, mediates endoplasmic reticulum stress-induced Caspase 4 activation and apoptosis [46]. Unlike GbTMEM214s, which have only one TMEM domain, the animal TMEM214 protein contains two transmembrane domains at its C terminus and a large at N terminus, and these domains are essential for the function of TMEM214. Nevertheless, current understanding of TMEM in animals is still not sufficient, and plant immunity is quite different from animal immunity, which limits the implication of the functional mechanism of GbTMEM214 from animal studies. Our study provides an important insight into the involvement of *GbTMEM214s* in plant-disease resistance, but the molecular mechanism remains to be revealed through further experiments.

## 4. Materials and Methods

### 4.1. The Bioinformatics Analysis of TMEM214 Superfamily

The genome data were downloaded at CottonFGD (https://cottonfgd.org/ accessed on 8 May 2020). The TMEM214 proteins in cotton were predicted with hidden Markov model (HMM) and HMMER 3.0 software. The HMM seed file of TMEM214 (PF10151) was obtained from the Pfam database (http://pfam.sanger.ac.uk/ accessed on 20 September 2021). The protein sequences of TMEM214 in other plants were obtained from NCBI (https://www.ncbi.nlm.nih.gov/ accessed on 20 September 2021). The sequence alignment of TMEM214 proteins were performed with ClastalX 1.83 software. The results were visualized with GENDOC software. The maximum likelihood method in MEGA 5.05 software was applied to build the phylogenetic trees of *TMEM214* genes. The ORFs were predicted with the Fgenesh subroutine in MolQuest 2.3.3 software. The gene structure was analyzed with GSDS 2.0 (http://gsds.cbi.pku.edu.cn/ accessed on 13 May 2022).

### 4.2. Protein Structure Analysis of TMEM214s

The domains in TMEM214 proteins were identified using the Simple Modular Architecture Research Tool (SMART, http://smart.embl.de accessed on 13 May 2022) following the instruction. Homology modelling of protein structures was applied to construct the 3D structural models using the SWISS-MODEL web server (https://swissmodel.expasy.org/ accessed on 13 May 2022).

### 4.3. Plant Materials and Treatments

VW-resistant variety Hai7124 and susceptible variety TM-1 were used in this study. Cotton seedlings were grown in a greenhouse at 28 °C during the day/night period of 16 h/8 h for 2 weeks. The *V. dahliae* strains were cultured on potato dextrose agar medium (PAD) at 24 °C for 5 days, in Czapek’s medium at 25 °C for 5 days, and then adjusted to 1 × 10^7^ conidia/mL with deionized water for inoculation. The seedlings were inoculated with *V. dahliae* using the dip-inoculation method, and the root samples were harvested at 0, 24, 48, 96 and 144 h, respectively.

### 4.4. RNA Isolation and Expression Pattern Analysis

Total RNA was extracted from root samples using (Omega Bio-tek) according to the manufacturer’s instructions, and cDNA was synthesized using PrimeScript RT reagent Kit with gDNA Eraser (TaKaRa). Quantitative PCR primers of *GbTMEMs* and *histone 3* (AF024716) were designed using Beacon Designer 7.0 software or following the previous studies (Appendix A), in which *histone 3* (*His3*) was used as a reference gene. Quantitative PCR was performed using TB Green qPCR Master Mix (TaKaRa) on the ABI QuantStudio 5 PCR System. The relative expression level was calculated as 2^−ΔΔΔCt^; ΔΔΔCt = [(Ct*_Gene_* − Ct*_His3_*)_x h_ − (Ct*_Gene_* − Ct*_His3_*)_0 h_]_Treatment_ − [(Ct*_Gene_* − Ct*_His3_*)_x h_ − (Ct*_Gene_* − Ct*_His3_*)_0 h_]_CK_.

### 4.5. Cloning of the GbTMEM214s in Hai7124

The gene-specific primers of *GbTMEMs* were designed using Primer Premier 5 software (Appendix A). Standard PCR reactions were performed using Ex Taq Hot Start Version DNA Polymerase (TaKaRa) to amplify the *GbTMEMs* with complete ORFs. The final product was cloned into pMD19-T Vectors (TaKaRa) and transformed into the *E.coli* strain DH5α.

### 4.6. VIGS Experiments

VIGS was performed using the binary pTRV1 and pTRV2 vectors to silence *GbTMEMs*. TRV2: *GbTMEMs* and TRV2: *GbCLA1* vectors were constructed by inserting the 3′UTR region specific sequences into pTRV2, and were subsequently transformed into the *Agrobacterium tumefaciens* (*A. tumefaciens*) strain GV3101. The *A. tumefaciens* cultures were grown overnight at 28 °C on a solid LB medium (50 µg/mL kanamycin, 25 µg/mL gentamicin), and then inoculated into a liquid LB medium (50 µg/mL kanamycin, 25 µg/mL gentamicin, 10 mM MES and 20 µM acetosyringone). The cultures were grown overnight in a 28 °C shaker. Cells were then harvested and re-suspended in an infiltration medium (10 mM MgCl2, 10 mM MES and 200 µM acetosyringone). The cell suspensions were adjusted to an O.D. of 2.0 and incubated at room temperature for 3 h. The *A. tumefaciens* cells containing the pTRV1 and pTRV2 vectors were mixed at a ratio of 1:1 and infiltrated into two fully expanded cotyledons of 2-week-old cotton plants. Thirty plants per VIGS treatment were infected with *V. dahliae* and a disease index was calculated as previously descripted [1].

### 4.7. Subcellular Localization of GbTMEMs

The ORFs of *GbTMEMs* were inserted into the pBin-GFP4 vector to express GbTMEMs-GFP fusion protein. The vector expressing the AtPIP2A-mCherry fusion protein was used as a positive control [47]. The vectors were then transformed into the *A. tumefaciens* strain GV3101, and the *A. tumefaciens* cells containing each vector were mixed at a ratio of 1:1 and infiltrated into *Nicotiana benthamiana* (*N. benthamiana*) leaves to transiently co-express fusion proteins following previous study [48]. Visualization of fluorescence signals was observed using a confocal laser scanning microscope (Zeiss, EVO-LS10) 3 d after infiltration.

## Figures and Tables

**Figure 1 plants-11-02342-f001:**
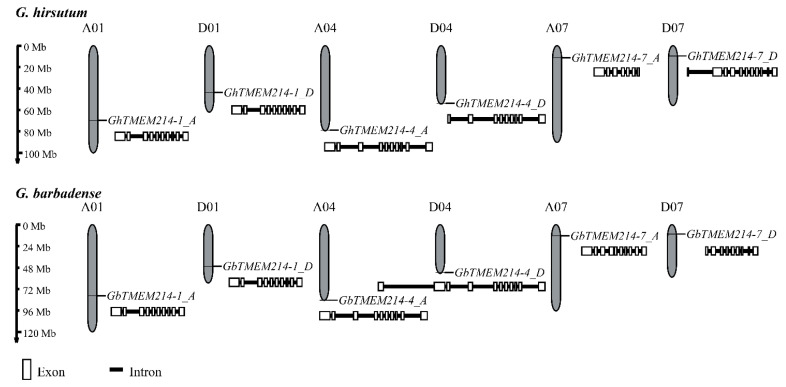
The position of *TMEM214* genes in the chromosomes of *G. barbadense* and *G. hirsutum.* The *TMEM214* genes in cotton were predicted with the hidden Markov model (HMM) and HMMER 3.0 software with the HMM seed file of TMEM214 (PF10151). *GbTMEM214s* and *GhTMEM214s* were named according to their localization on the chromosomes of *G. barbadense* and *G. hirsutum*, respectively. The gene structure was analyzed with GSDS 2.0 (http://gsds.cbi.pku.edu.cn/ accessed on 13 May 2022). The exons were boxed and the introns were lined.

**Figure 2 plants-11-02342-f002:**
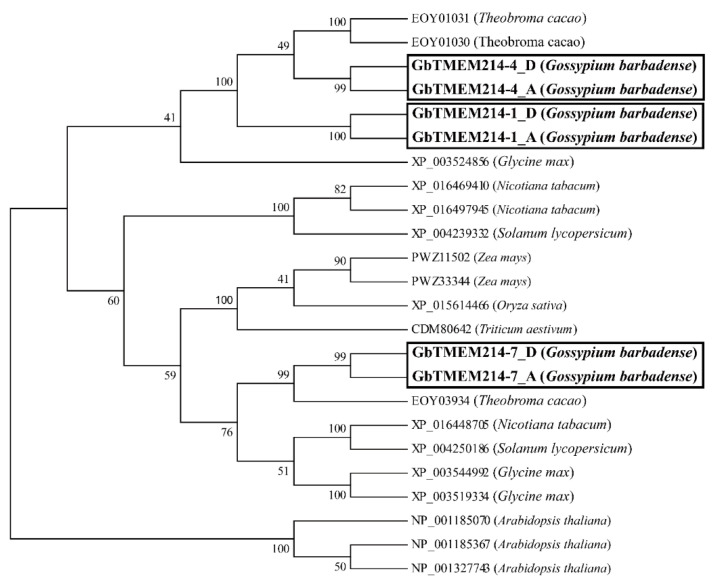
The phylogenetic trees of TMEM214 proteins in plants. MEGA 5.05 was applied to construct the phylogenetic trees of TMEM214 proteins. GbTMEM214s from *G. barbadense* were bolded and marked with a box.

**Figure 3 plants-11-02342-f003:**
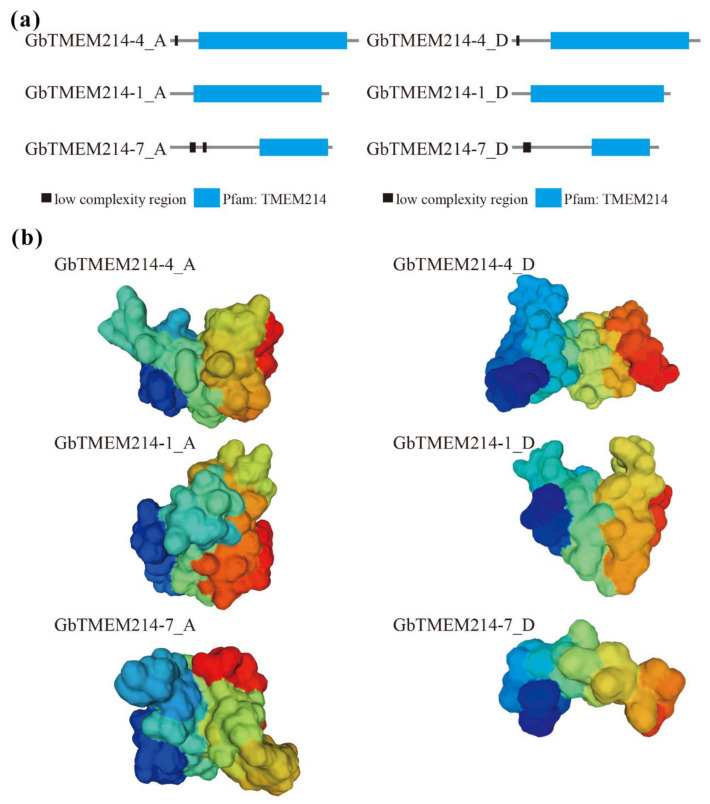
The protein structure of GbTMEM214s. (**a**) Protein sequence of GbTMEM214s in D-genome were analyzed with the Simple Modular Architecture Research Tool (SMART, http://smart.embl-heidelberg.de/ accessed on 13 May 2022). The structural models automatically generated with the website were recolored to highlight each domain. The low complexity region and TMEM214 domain were represented by black and blue boxes, respectively. (**b**) 3D structural models of GbTMEM214s D-genome. Homology modeling of protein structures was applied to construct the 3D structural models using the SWISS-MODEL web server (https://swissmodel.expasy.org/ accessed on 13 May 2022). The optimal PDB modeling templates were selected according to the Global Model Quality Estimate (GMQE) value (Appendix A).

**Figure 4 plants-11-02342-f004:**
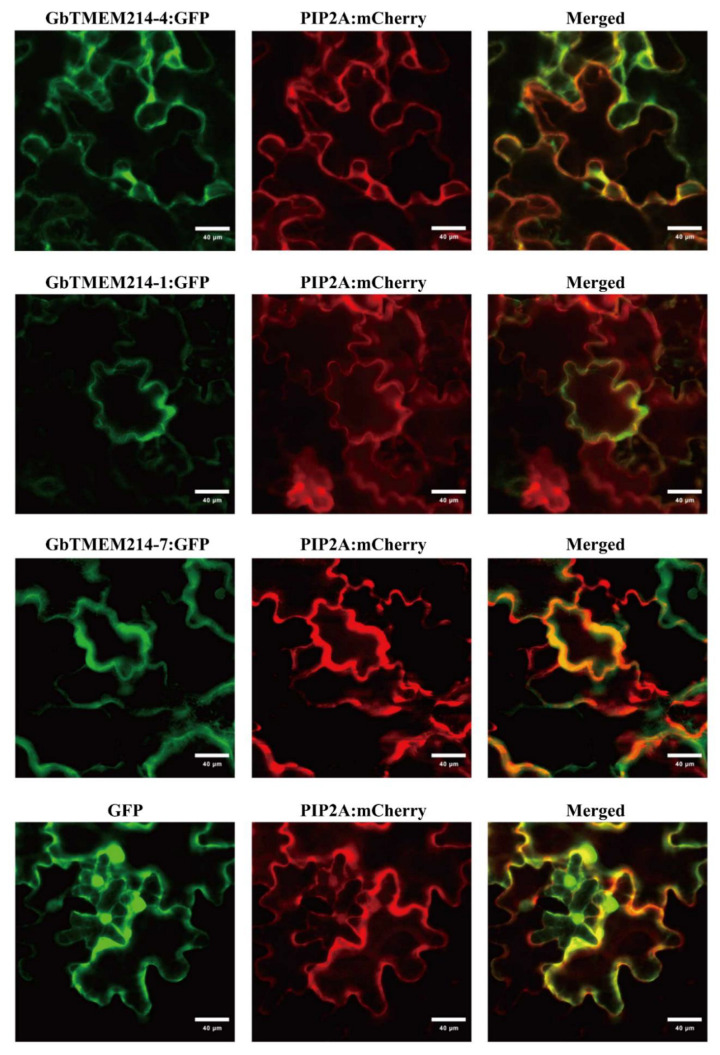
The subcellular localization of GbTMEM214s. The subcellular localization of GbTMEM214s-GFP in tobacco leaves. The constitutive GFP serves as a control. AtPIP2A-mCherry was used as a membrane-localization marker. The constructs were transiently expressed in *N. benthamiana* leaves. The green signal of GFP was fused with the red signal of membrane marker to determine the localization of the protein. Bars = 40 µm.

**Figure 5 plants-11-02342-f005:**
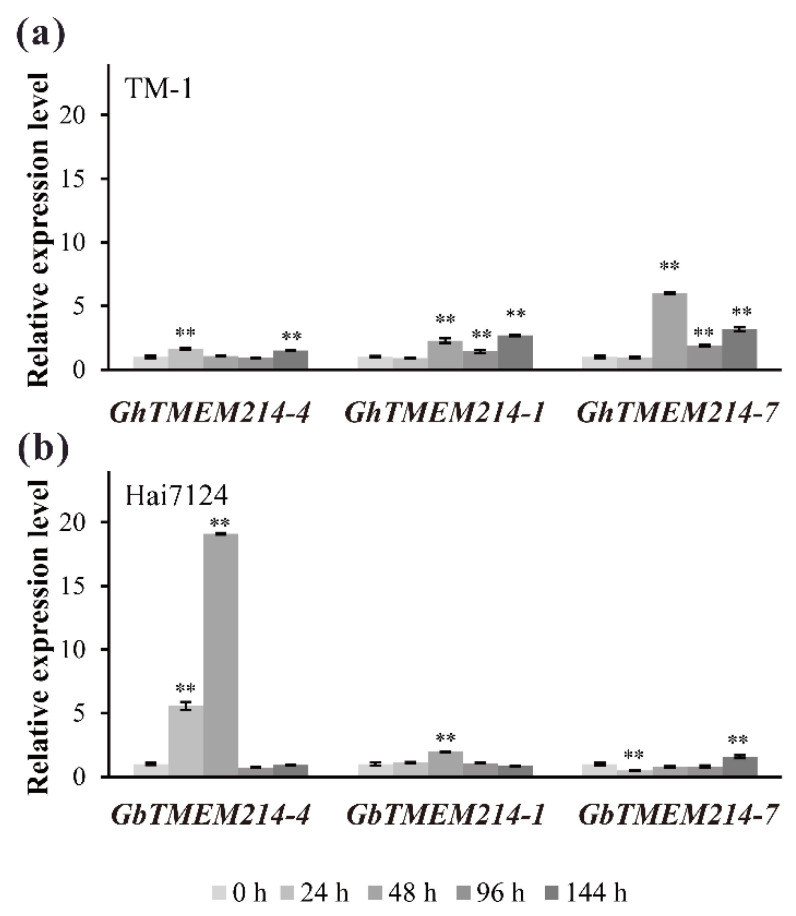
The relative expression levels of *TMEM214* genes under the challenge of *V. dahliae*. (**a**) *GhTMEM214* genes in susceptible variety TM-1; (**b**) *GbTMEM214* genes in resistant variety Hai7124. The qRT-PCR was applied to analyze the expression of *TMEM214* genes. The data were expressed relative to 0 h for each gene. “**” represent significant differences relative to each control and *p*-value < 0.01, based on Student’s *t*-test. Each value was the mean ± SD of three biological determinations.

**Figure 6 plants-11-02342-f006:**
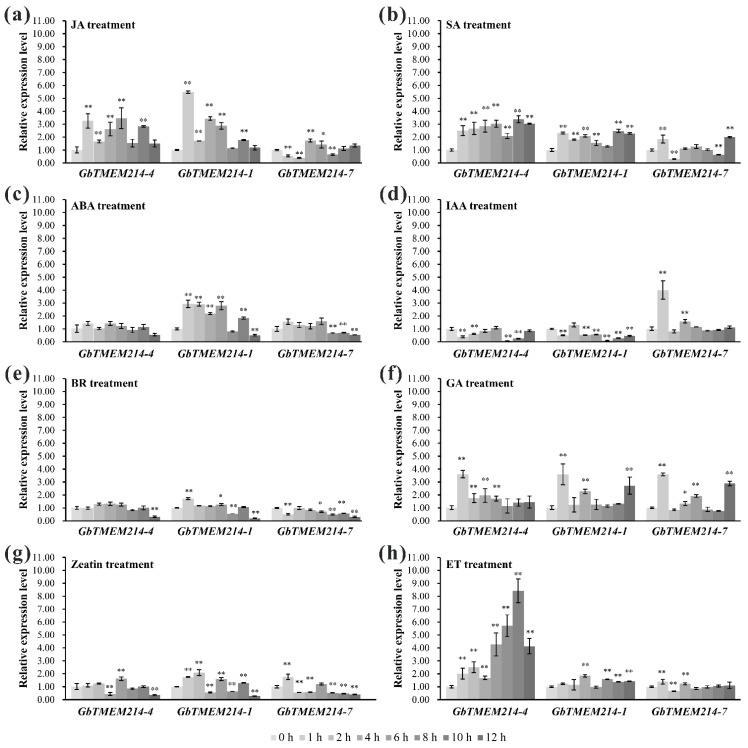
The relative expression levels of the *GbTMEM214* genes with the treatment of phytohormone. (**a**) jasmonic acid (JA) treatment; (**b**) salicylic acid (SA) treatment; (**c**) abscisic acid (ABA) treatment; (**d**) indole-3-acetic acid (IAA) treatment; (**e**) brassinosteroid (BR) treatment; (**f**) gibberellin (GA) treatment; (**g**) zeatin treatment; (**h**) ethylene (ET) treatment. The qRT-PCR was applied to analyze the expression of the *TMEM214* genes. The data were expressed relative to 0 h for each gene. “*”, “**” represent significant differences relative to each control and *p*-value < 0.05 or *p*-value < 0.01, based on Student’s *t*-test. Each value was the mean ± SD of three biological determinations.

**Figure 7 plants-11-02342-f007:**
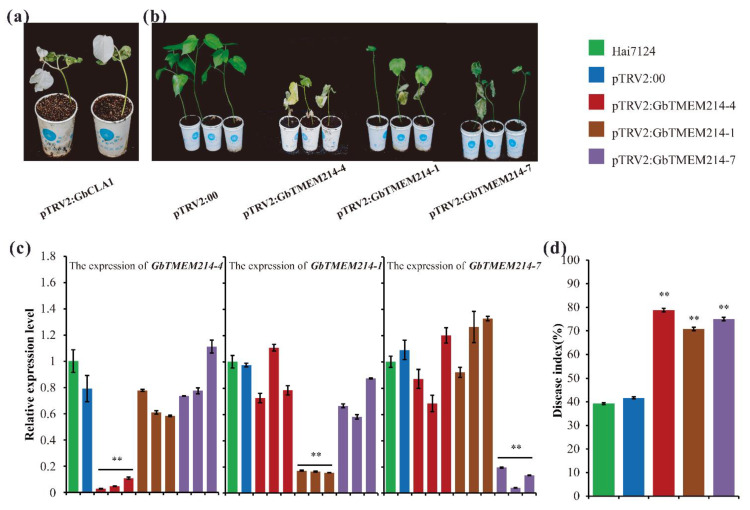
The resistance function analysis of the *GbTMEM214* genes using VIGS. (**a**) The cotton gene *GhCLA1* was used as a positive control, and VIGS of *GhCLA1* results in a phenotype of white leaves; (**b**) The phenotypes of Hai7124 under infection with *V. dahliae* after VIGS with *Agrobacterium* carrying *pTRV2:GbTMEM214s* and *pTRV2:00*, and the photos were taken at 42 days after *V. dahliae* inoculation; (**c**) qRT-PCR analysis of the expression levels of *GbTMEM214s* in the silenced lines; (**d**) The disease index of plants with silenced *GbTMEM214s*. The data for each cotton line were filled in different colors. The qRT-PCR data were expressed relative to Hai7124 for each gene. The results were evaluated at 28 d after *V. dahliae* inoculation, with three replications containing at least 20 plants each. “**” represent significant differences relative to each control and *p*-value < 0.01, based on Student’s *t*-test. Each value was the mean ± SD of three biological determinations.

## Data Availability

Not applicable.

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
