# Peer review of "Membrane Localized GbTMEM214s Participate in Modulating Cotton Resistance to Verticillium Wilt"

_plants, 2022, doi:10.3390/plants11182342_

Round 1
Reviewer 1 Report
Comments
1. Line 19-22, the author stated there were six genes, but only three genes were named. There is the same issue in Line 87-92.
2. Line 95-97, "The homologous genes of GbTMEM214-4 were different in the first exon of 3'UTR region, and similar differences existed among homologous genes of GbTMEM214-7." The authors did not show the differences between the 3'UTR region in Fig. 1.
3. Line 97-100, please explain why the author did not select G. raimondii and G. arboreum when constructing the phylogenetic tree.
4. Line 101-102, I suggest changing the sentence "Among TMEM214s in cotton, GbTMEM214-4 was closer to EOY01031 and EOY01030 in 101 cocoa, while GbTMEM214-7 was closer to EOY03934." to "Among TMEM214s in selected plants, GbTMEM214-4 was closer to EOY01031 and EOY01030 in 101 cocoa, while GbTMEM214-7 was closer to EOY03934."
5. Line 152-157, the name of genes in Fig. 5a should be G. hirsutum (TM-1), as mentioned in the caption, but the names in the figure were G. barbadense.
6. Line 159, why use phytohormone to induce GbTMEM214s? Is there any relevant literature?
7. Line 185-188, the authors should add the measurement approach of disease index in the M&M section.
8. Line 185-186, the sentence "The disease index was calculate 25 days after inoculation"; however, in Line 196-198, the sentence "The disease index of plants with silenced GbTMEM214s. The results were evaluated at 28 d after V. dahliae inoculation...", 25 days or 28days? The authors should confirm it.
9. Wrong labels of b, c, and d in Fig. 7.
10. Line 251-259, the parameters of HMM 3.0 software should be added, as well as the exact software used for phylogenetic tree construction.
Reviewer 2 Report
In this manuscript, Zhao et al. sought to characterize a family of transmembrane proteins, the TMEM214s, with regards to its contribution to Verticillium wilt in Gossypium barbadense. They cloned three TMEM214 homologues from each A/D sub-genome of G. barbadense and used transient co-expression assays to suggest that these GbTMEM214 proteins are membrane-localized. They also monitored the expression of GbTMEM214 genes in response to Verticillium dahliae infection as well as treatment with various phytohormones. Finally, they used VIGS to knock down the expression of GbTMEM214 genes and observed increased susceptibility to V. dahliae in the silenced plants. The manuscript is generally suitable for publication, although it would greatly benefit from the inclusion of additional information, as described below.
Major Comments:
What is the percent identity of the DNA/protein sequences for these genes when you compare the A and D subgenome homologues to each other? Presumably the sequences could not be resolved for the qRT-PCR.
Does the sentence starting at line 148 refer to comparisons of basal levels of gene expression? If not, were there any differences in basal gene expression for these genes when comparing either within or between the two varieties of cotton?
In Figure 3, protein structures were predicted by homology modeling. With this approach, similar sequences are often mapped onto the same predicted structure, so the prediction of different structures suggests that the GbTMEM214 sequences are quite divergent. Again, it would be very helpful to know the percent amino acid identity among the six GbTMEM214 proteins. Please also indicate the specific protein and PDB file used to generate each homology model. What percent of the protein sequence is covered by the models? If there is partial coverage, are the models based on different regions of the GbTMEM214 protein, or do they all encompass roughly the same region?
The disease phenotypes were evaluated at 28 – 42 d after inoculation; at these timepoints, were any uninoculated plants monitored to check for any phenotypic differences between pTRV2:00 plants and GbTMEM214-silenced plants?
Figure 7: It is unclear whether or not the specificity of the silencing constructs was assessed. For example, is the expression of GbTMEM214-4 altered in a GbTMEM214-1-silenced plant? The authors only make general conclusions about the impact of GbTMEM214 silencing, and do not make specific claims about any one gene, so this oversight does not invalidate their conclusions, but it would have been nice to see some controls for silencing specificity.
Minor Comments:
Figure 4: Please note in the legend that the constructs were transiently expressed.
Figure 5: Please state that the data are expressed relative to t0 for each gene. Also, the time points in the figure axis are a mix of ‘h’ and ‘H’, and should be made consistent, as in Figure 6.
The figure labelling in Figure 7 does not match the legend (i.e. (c) and (d) are in the legend but not the figure, and (b) refers to images rather than the graph).
Line 41: “silent expression” should be “silencing”
Line 62: Reference formatting
Line 69: I think you mean “phospholipid bilayer” instead of “biofilm”, which usually describes microbial communities
Line 79: “be participated” should be “participate”
Line 105: Spell out hidden Markov model (HMM) the first time that it is used
Line 132: Please describe how the structural models were generated.
Line 141: To help the reader, please indicate that TM-1 and H7124 are susceptible and resistant cotton varieties, respectively.
Line 174: “Bbrassinosteroid” should be “Brassinosteroid”
Line 184: italicize V. dahliae
Line 268: For consistency, the variety name Hai7124 should be used throughout the manuscript instead of H2174.
Line 290: CYP72A is not mentioned anywhere else in the manuscript, so this seems to be a typo.
